# Photoluminescence and Fluorescence Quenching of Graphene Oxide: A Review

**DOI:** 10.3390/nano12142444

**Published:** 2022-07-17

**Authors:** Xinzhe Xiao, Yumin Zhang, Lei Zhou, Bin Li, Lin Gu

**Affiliations:** School of Chemical Engineering and Technology, Sun Yat-Sen University, Zhuhai 519082, China; xiaoxzh5@mail2.sysu.edu.cn (X.X.); zhangym236@mail2.sysu.edu.cn (Y.Z.); zhoul8@mail.sysu.edu.cn (L.Z.)

**Keywords:** graphene oxide, quantum dots, photoluminescence, fluorescence quenching

## Abstract

In recent decades, photoluminescence (PL) material with excellent optical properties has been a hot topic. Graphene oxide (GO) is an excellent candidate for PL material because of its unique optical properties, compared to pure graphene. The existence of an internal band gap in GO can enrich its optical properties significantly. Therefore, GO has been widely applied in many fields such as material science, biomedicine, anti-counterfeiting, and so on. Over the past decade, GO and quantum dots (GOQDs) have attracted the attention of many researchers as luminescence materials, but their luminescence mechanism is still ambiguous, although some theoretical results have been achieved. In addition, GO and GOQDs have fluorescence quenching properties, which can be used in medical imaging and biosensors. In this review, we outline the recent work on the photoluminescence phenomena and quenching process of GO and GOQDs. First, the PL mechanisms of GO are discussed in depth. Second, the fluorescence quenching mechanism and regulation of GO are introduced. Following that, the applications of PL and fluorescence quenching of GO–including biomedicine, electronic devices, material imaging–are addressed. Finally, future development of PL and fluorescence quenching of GO is proposed, and the challenges exploring the optical properties of GO are summarized.

## 1. Introduction

Graphene is an advanced two-dimensional (2D) carbon material with thickness of a single atomic layer. Since it was first discovered in 2004, it has rapidly attracted wide attention [1,2]. As the origin of 2D materials, the intriguing physical properties of graphene and other 2D materials have a wide and promising application potential in many fields of composite materials [3], biomedicine [4], anti-corrosion coatings [5,6], electronic devices [7], infrared detection [8] and so on. Graphene oxide (GO), as a product of graphite oxidation, has more active groups than graphene, due to the increase of oxygen-containing functional groups (e.g., -C-O, -C=O, -COOH) in the oxidation process. To date, the research on GO has surpassed that of its parent material [9]. In the research on GO, an array of unanticipated photophysical structures and subsequent luminescence properties have emerged through the regulation of the proportion and size of sp^2^ and sp^3^ carbon atoms in different concentrations, which has opened a couple of potential research areas. One of the major topics regarding GO is the photoluminescence (PL) and quenching of GO, its quantum dots (GOQDs), and graphene quantum dots (GQDs) [10]. The PL is theoretically impossible for graphene material to achieve due to the zero band gap [11]. However, with the insertion of multiple oxygen-containing functional groups into GO and GOQDs, PL can be realized over a large wavelength range by modulating the band gap. In addition, it was found that the PL of GO and GOQDs can be quenched under certain circumstances, and other fluorescent substances could be quenched by GO as well.

The PL and fluorescence quenching mechanisms of GO, GOQDs, and GQDs in the ultraviolet and visible light range are still ambiguous so far, although some possible mechanisms were proposed based on their experimental phenomena and density functional theory (DFT) calculations [12]. In addition, clarifying PL and fluorescence quenching mechanisms plays a vital role in further studying carbon materials and expanding their application prospects. To address the aforementioned issues, this paper will comprehensively review the following aspects in detail shown in Figure 1: luminescence mechanism, luminescence influencing factors and regulation, luminescence characterization methods, fluorescence quenching mechanism and regulation, GO luminescence application, and fluorescence quenching application.

## 2. Luminescence Mechanism

Graphene is composed of sp^2^ hybrid carbon atoms in its 2D layered structure. In its band structure, the top of the valence band and the bottom of the conduction band intersect at the Dirac point in the Brillouin zone, which has the characteristics of a zero band gap. Thus, it is nearly impossible to directly observe the PL of graphene [13]. The band gap can be obtained by graphene through quantum confinement, chemical modification, or hybridization [14], among which the chemical modification is the most widely adopted method. For example, graphene is oxidized to GO, with many oxygen-containing functional groups introduced, such as -C-O, -C=O, -COOH, leading to a stronger confinement effect. The existence of defects on the surface of GO and smaller sp^2^ clusters will bring a considerable band gap [15], which results in the separation of the valence band and conduction band. In fact, the carbon atoms with sp^3^ are introduced into the original sp^2^ as defects during the reduction process [16]. In 2008, Dai et al. reported for the first time the PL phenomena of nano graphene oxide (NGO) and polyethylene glycol modified NGO (NGO-PEG) in the visible and infrared regions. (Figure 2a) [17]. Under the excitation of 400 nm, the emission position of GO was about 570 nm, and, for NGO-PEG, the maximum emission blue-shifted to roughly 520 nm (Figure 2b), which may be due to the reduction of the size of the GO sheet by chemical activation and PEGylation.

### 2.1. Common Reduction Methods

It is mentioned above that the oxidation of graphene to GO via chemical modification can introduce the band gap, and the appropriate reduction of GO by different methods to obtain reduced graphene oxide (rGO) can effectively adjust the band gap of graphene, thus realizing the tunable optical properties of GO. Essentially, the biggest difference between GO and rGO is the degree of oxidation [18]. In the work of Al-Gaashani et al., the XPS of GO and rGO obtained by different methods was studied in detail [19]. The C/O ratios of GO obtained by three different methods ranged from 1.63 to 2.77, and the C/O of rGO obviously increased, further illustrating the essential difference between GO and rGO. In short, it is necessary to review and understand the reduction methods of GO as well as the luminescence mechanism of GO and GOQDs.

The annealing of GO is a simple heat treatment procedure that can achieve GO reduction. Chen et al. enhanced the fluorescence properties of GO by annealing NGO under certain conditions (at 80 °C for 9 days) [20]. The results revealed that the annealing process can keep the content and distribution of oxygen functional groups in the GO structure efficiently, so that the 2D plane of GO contains both sp^2^ and sp^3^ clusters simultaneously, but the annealing time is longer. According to the confocal imaging results, the green fluorescence signal on the annealed NGO interface is significantly higher than that on the GO interface, and the fluorescence intensity of NGO is ~2.5 times higher than that of GO. Kumar et al. adopted a mild thermal annealing process, which did not involve chemical treatment. This is also the first detailed study of the PL emission transition of GO from the ultraviolet range to the blue range during thermal reduction [21]. Tang et al. prepared fluorinated reduced graphene oxide (F-rGO) by annealing reduced graphene oxide (rGO) and xenon fluoride (XeF), and investigated its structure, band gap, and optical properties. A series of characterization results clearly display that the CF_2_ group has affected a substantial number of vacancies in the structure of F-rGO, and the existence of these vacancies is conducive to the opening and adjustment of the band gap [22]. The band gap can be enhanced from 0.42 eV to 3.0 eV through the preparation process by raising the fluorine content from 0.29 at% to 0.98 at%. Wang et al. also reduced GO by a simple and effective two-step fluorination method to explore its efficient PL, and the relevant test results show that its fluorescence quantum yield (PLQY) was as high as 66%, which has been the highest value of PLQY in various modification or reduction methods of GO [23]. Meanwhile, different types of fluorine-related groups in F-rGO have diverse fluorine effects in boosting PL. They found that during the excitation process, the coexistence of fluorine and aromatic regions in F-rGO leads to a fresh electronic band gap structure, resulting in a large number of stimulated electrons, thereby finally improving their PL properties.

Maiti et al. gradually reduced graphene oxide (GO) by infrared irradiation and realized the optical properties of tunable GO [24]. With the adding of infrared exposure time and irradiation power density, the yellow-red emission peak of GO gradually decreased and blue-shifted, while the blue PL emission peak became the prominent peak of continuous reduction. In addition, laser reduction is an efficient method with flexible operation and no chemical corrosion. It also shows an excellent reduction effect for other materials besides GO. Jahanbakhshian et al. successfully prepared rGO suspensions with different reduction degrees with 405 nm continuous wave laser irradiation, and studied the linear and nonlinear optical properties of GO and rGO samples with a 532 nm continuous wave laser beam measurement [25]. Gomaa et al. successfully reduced GO films using a laser diode with a wavelength of 808 nm at room temperature and air [26]. The maximum PL peak blue-shifted from 587 nm to 568 nm after a three-minute reduction.

Hasan et al. treated GO with ozone to achieve the purpose of changing the optical band gap [27]. The underlying mechanism for the change of the band gap is that the ozone-induced functionalization reduces the size of the sp^2^ cluster, thereby affecting the GO band gap and emission energy. Kumar et al. used three different methods to reduce the produced GO to rGO–green chemical reduction of vitamin C, hydrothermal reduction techniques, and thermal reduction techniques–and compared these three methods [28]. Caballero-Briones et al. added cesium hydroxide (CsOH) solution to the GO suspension. Cs-modified reduced graphene oxide (CsRGO) was synthesized, and its PL spectrum was studied. The emission at 461 nm in the PL spectrum was ascribed to the π-π* transition of the sp^2^ carbon atoms [29].

Umadevi et al. employed the dielectric properties of ionic liquid crystal (ILC) to promote fast-speed microwave heating to prepare rGO. The reduction of GO by ILC exhibits a better effect compared with other reduction methods [30]. Compared with the several aforementioned reduction methods, the electrochemical reduction has the merits of environmental protection, energy-saving, and excellent controllability. It has gradually become the prevalent reduction method for researchers. Bai et al. summarized the electrochemical reduction of GO to change the band gap of GO, and some shortcomings of electrochemical reduction methods were also mentioned; for instance, the reduction reaction only occurs on the surface of the electrode, and the reduction capacity is restricted by multiple factors such as the electrode area [31]. Ménard et al. prepared rGO using the spray pyrolysis deposition method, which also shows a high degree of reduction [32]. Figure 3 displays the schematic diagram of the partial reduction of GO. Although it is remarkable that the above reduction methods have ideal reduction effects, they may change the original properties of GO irreversibly [33].

### 2.2. Luminescence Mechanism of GO

Many studies have noticed that the PL of GO mostly consists of two bands: one is in the blue range (blue band), while the other is in the 500~650 nm range (long wavelength band). However, the PL mechanism of GO is still ambiguous. Clarifying the luminescence process is critical for understanding and regulating the optical characteristics of GO and will expand the design of graphene-based materials.

Chhowalla et al. deemed that the π-π* transition of sp^2^ clusters isolated by sp^3^ matrix in the plane after reduction of GO by hydrazine is the dominant factor of blue band luminescence or enhanced PL (Figure 4a) [34]. I, II and III correspond to GO structural models of different reduction stages (reduction degree gradually increased). In the atomic structure, the origin of GO luminescence comes from the bottom and vicinity of the conduction band to the electron-hole recombination in a wide range of valence bands, and the size of the sp^2^ clusters determines the local energy gap. In essence, the fluorescence emission of GO is due to the behavior of excitons (electron-hole pairs), according the most widely accepted theory, which is currently the basis of many PL studies on GO [35]. Galande et al. proposed that the PL of GO originates from the quasi-molecular fluorescence group, and thought that the three functional groups of -C-O, -C=O and -COOH are related to the PL of GO, because they observed that -COOH has electronic coupling with the atoms near the graphene sheet (G) [36]. In addition, they found that the blue band dominates when pH > 8, and attributed this to the optical transition from (G-COO^−^)* to G-COO^−^. On the other hand, the long-wavelength band becomes obvious when pH < 8, which is attributed to the optical transition from (G-COOH)* to (G-COOH), as -COOH was protonated into carboxylic acids.

Moreover, the free sawtooth position similar to the triplet state is also theoretically expected and experimentally confirmed to be the possible mechanism of PL of chemically derived GO (Figure 4b,c) [37,38]. Chien et al. proposed that PL is induced by the disorder-induced localized state [39]. They found that GO in water exhibited a process of multi-exponential decay kinetics from 1 picosecond (ps) to 2 nanoseconds (ns) by a time-resolved fluorescence measurement. It was found that fluorescence was mainly due to the electronic transition between the boundary of the non-carbon oxide region and the carbon oxide atomic region, according to the concept of molecular orbital theory. In addition, chemical reduction is also considered to be one of the reasons for PL of GO; the mechanism diagram they proposed is displayed in Figure 4e [40]. Chu et al. deemed that the strong emission in the blue band is related to the carbon defect state formed during the reduction process, and the long-wavelength PL enhanced by functionalized rGO is generated by the sp^2^ cluster size effect [40]. He et al. studied the Mn^2+^-mediated energy transfer process as a multifunctional source of PL in GO (Figure 4f) [41]. They found that the excitation-independent blue emission and the two fixed PL peaks near 330 nm and 450 nm were related to the Mn^2+^-mediated energy transfer process.

### 2.3. Luminescence Mechanism of GOQDs

Quantum dots are a kind of material with nanoscale in three-dimensions, which have a high quantum confinement effect due to their tiny sizes. These distinctive effects also have an impact on their optical qualities. Usually, there are two principal ideas for preparing fluorescent quantum dots: “top-down” from different carbon sources and “bottom-up” from small molecules or polymers. Cutting GO by reduction or oxidation is a popular method to prepare quantum dots GOQDs. However, the PL mechanism of GOQDs is also ambiguous like GO.

In Section 2.2, it is mentioned that the sp^2^ cluster in the GO plane is the main source of PL, and the size of the sp^2^ cluster will affect PL. Meanwhile, the size of the sp^2^ cluster depends on the size of GOQDs, thereby affecting the band gap and even the emission wavelength. Nevertheless, the theory does not seem to be applicable for GOQDs, because it was found that PL quantum dots that are both size-dependent and size-independent can be synthesized [34,42].

Lau et al. studied the PL origin of GOQDs, and thought that the unique optical properties of GOQDs were attributed to the self-passivation layer formed by the surface state between the π* band and the π level (Figure 5) [43]. Rao et al. prepared GOQDs in two different solvents. The PL spectra of these samples show that functional groups did not contribute to PL [44]. In addition, it was observed that PL appeared to originate from the edge state of GOQDs, and the edge was passivated due to annealing, which led to the quenching of PL signals. Finally, they proposed that quantum dots could be regarded as a highly conjugated benzene ring system. Its emission peak comes from HOMO-LUMO energy levels similar to aromatic molecules.

Yang and co-workers deemed that the competition between the defect state emission and the eigenstate emission is considered to be the PL mechanism of GOQDs, and the energy level shift between the eigenstate and the edge state determines their optical properties [45]. If the energy level shift between the eigenstate and the edge state is large enough, the fluorescence is dominant; if the energy level offset is small enough, the long carrier lifetime in the eigenstate provides the possibility for the system to cross from the single excited state to the triple excited state of the edge state. Afterwards, Yang et al. found that the principal PL center of quantum dots contains the quantum confinement effect of conjugated π domains, the surface/edge state, and the cooperative effect of these two factors [46]. Behzadi et al. thought that the possible luminescence mechanism can be classified as the eigenstate emission and the non-eigenstate emission. The intrinsic emission is caused by the smallest oxidized sp^2^ cluster or size effect [47]. On the other hand, the external emission comes from the surrounding functional groups, including hydroxyl, epoxy, carbonyl, and carboxyl groups, which are formed on the edge and base.

## 3. Influence Factors and Regulation of Luminescence

The PL of GO, GOQDs, and GQDs is affected by many factors, and the effective regulation of PL can be realized by clarifying its influencing factors. It is found that the influencing factors of PL include temperature, pH, size, elements, oxygen-containing functional groups, and some unconventional regulatory factors, according to previous studies [48,49,50,51]. The following will discuss the influencing factors on PL.

### 3.1. Temperature

The temperature dependence of GO has long been reported in the literature [52]. He et al. observed two obvious PL bands in the study of GO temperature dependence, with 3.18 eV (A band) and 2.53 eV (B band) as the centers (Figure 6a) [53]. The energy band A gradually became insignificant as the temperature increased, and the energy band B dominated PL. Figure 6a (I–V) displays five typical PL spectra recorded at 13, 150, 200, 240, and 300 K, respectively. It is evident that the evolution of PL with temperature indicates the competition between the two bands (band A and band B). Simultaneously, the integral intensities of bands A and B at different temperatures were extracted and displayed in Figure 6b,c. In the whole temperature range, the intensity of band A showed an overall decreasing trend, except for a slight increase from 140 to 180 K, which is similar to the temperature-dependent band gap contraction of semiconductors, indicating the quasi-band-to-band transition property of band A. If a large number of defects strongly interfere with the GO plane, electron jumps will occur in the sp^2^ cluster at low temperature, which is related to the defect freezing effect. He et al. attributed band A to the exciton hopping process between sp^2^ clusters. Contrary to band A, band B was strongly suppressed at low temperatures. The strength of band B remained almost constant below 150 K, and then it was observed to rise sharply. The strength reached a peak at roughly 240 K, and then dropped sharply (Figure 6c). It is worth noting that it showed a typical negative thermal quenching (NTQ), and the energy of band B remained almost constant throughout the temperature range. This means that the emission of band B comes from the deep localized state. This is probably due to sp^3^ defects in the π-π* band gap.

Cuong et al. found that the two different reduction methods, thermal reduction of graphene (TRG) and chemical reduction of graphene oxide (CRG), were temperature-dependent (Figure 6d,e) [54]. When the temperature increased from 10 K to 300 K, the peak position of the CRG sample shifted blue with the fluctuation of temperature, while the peak position of the TRG sample remained almost unchanged. The results showed that the carrier localization effect in the sp^2^ cluster played a dominant role in the emission peak shift, and the temperature-induced band gap shrinkage effect led to the red shift of the PL peak. In addition, the peak of CRG film was located at a lower wavelength than that of TRG film at room temperature (300 K). Cuong et al. thought that this is because TRG has a larger sp^2^ size compared with CRG, and the relatively weak carrier limitation leads to a smaller band gap.

Saha et al. used 2,6-diaminopyridine (DAP) to functionalize GO (DAP-fGO), in order to obtain excellent optical properties, and the temperature-dependent PL measurements of DAP-fGO films prepared on quartz plates were carried out at temperatures ranging from 78 to 300 K using a low temperature optical thermostat (Figure 7b), and the temperature-dependent PL lines from 78 K to 300 K are represented by lines of different colors in Figure 7b. The hybridization energy levels due to the functionalization of GO were clarified [55]. The PL spectra related to the excitation wavelength are shown in Figure 7c, which demonstrates that a third peak was observed in the red region with the increase in the excitation wavelength, and black, blue and red lines represent the spectra excited at 340 nm, 420 nm and 480 nm respectively. When the DAP-fGO film was excited at the wavelength of 420 nm, a strong peak was observed in the green region (530 nm), and a tail was observed in the red region (650 nm). When the excitation wavelength was 480 nm, this tail became more obvious. Therefore, it was confirmed that the emission occurred in blue, green, and red regions by measuring the temperature and excitation-related PL of the DAP-fGO sample.

### 3.2. pH

When GO and quantum dots are dispersed in different pH conditions, their optical properties will also change, which is also conducive to the study of the contribution of different groups to the fluorescence emission. Ouyang et al. reported on the relationship between the PL spectra of GO and the conditions of the aqueous solution, such as pH value and concentration, and found that when the pH value is low or the concentration of GO is high, the PL in the visible range becomes obvious [56]. Meanwhile, the lifetime of PL in the visible range is longer than that in the ultraviolet range via time-resolved PL spectra. Pan et al. proved that blue fluorescence from GQDs is pH-dependent [38]. At high pH levels, the fluorescence is strong enough to be observed by the naked eye, while at low pH conditions, the fluorescence is almost quenched. They thought that the protonation of the emission zigzag site of the ground state of σ^1^π^1^ under acidic conditions leads to fluorescence quenching, and the fluorescence recovery is achieved through deprotonation under alkaline conditions.

Zou et al. studied the PL of GO dispersions with different pH values of 2, 4.5, 7.5, 10, and 11.5 [57]. Figure 8 presents a 2D luminescence diagram of GO dispersions, which is treated by ultrasound with different energies up to 20 MJ·g^−1^ at pH values of 2 to 11.5. Halder et al. used anionic surfactant (SDS) to regulate the PL of GO in different pH environments [58]. When pH ≈ 2, the surface micelles of GO sheets formed, resulting in a nonpolar environment around the fluorescent groups of GO, which hindered the solvent relaxation. As a result, the PL band showed an abnormal 36 nm blue shift. However, at pH ≈ 10, due to the negative-charged -COO^-^ at the edge of GO, SDS experienced multiple interactions with GO sheets. The rejection between the negative-charged GO sheets and the embedding of SDS in the GO substrate weakened the π-π stacking interaction and changed its electronic environment, thus forming many separated GO layers, resulting in an enhanced PL intensity at 303 nm.

### 3.3. Size

It is mentioned in Section 2.2 that the band gap between sp^2^ carbon atoms is one of the possible reasons for PL. Obviously, appropriate size will be one of the factors affecting its PL, because size can change the sp^2^ region, thereby changing the band gap. Li et al. synthesized high PL GOQDs via a simple one-pot hydrothermal method, and then separated them with dialysis bags of different molecular weights to obtain four separated GOQDs with different sizes [50]. The study found that four separated quantum dots showed different PL intensities. The intensity of the emission peak becomes stronger as the size of the separated quantum dots decreased (Figure 9). Finally, the smallest size quantum dots show the strongest PL intensity among the four separated quantum dots. Matsuda et al. reported that the PL of GOQDs separated by size exclusion high performance liquid chromatography changed dramatically from ultraviolet to red [59]. The 2D PL of GOQDs showed four different emission peaks at 330 nm, 440 nm, 520 nm, and 600 nm. The main luminescence characteristics of the separated GOQDs showed discrete changes depending on the overall size of GOQDs. This indicates that PL changes occur due to the differences in the density, shape, and size of the available sp^2^ fragments in GOQDs.

### 3.4. Elements and Oxygen Functional Groups

The doping of different elements and the presence of oxygen-containing functional groups will also affect PL [51,60,61]. The doping of some elements promotes luminescence, while the doping of some other elements has a significantly negative effect on luminescence.

Li et al. studied the PL behavior of GQDs doped with different elements (Figure 10a) [51]. The GQDs and GQDs doped with Cl, N, P, S were prepared by electrochemical method and hydrothermal method respectively (Figure 10b). It was found that the doping with Cl and N can form luminescence centers and improve the luminescence intensity of GQDs (Figure 10c,d). The luminescence intensity of Cl-GQDs increased by one time over the undoped GQDs. The doping of the S element forms a small amount of quenching centers, and the luminescence intensity is slightly lower than that of undoped GQDs. The doping of P forms a large number of quenching centers, resulting in almost no luminescence of P-GQDs (Figure 10e). Yang et al. also studied the effect of S and N doping and simultaneous doping on the fluorescence properties of GO (Figure 10f) [60]. N, S co-doping will result in different surface states of GO, and many of the captured electrons may have greater radiative recombination probability, resulting in higher fluorescence intensity than the original GO.

Kim et al. found that the red shift was observed in the emission spectra of GO with the increase of oxidation degree [61]. They thought that it was due to the increase of the concentration of oxygen-containing functional groups, resulting in the increase of sp^2^ cluster size, which highlighted the understanding of the effect of oxidative functional groups on the PL properties of GO. Cui et al. found that the position of hydroxyl on the base surface of GOQDs would accelerate the non-radiative decay, while the position of hydroxyl on the edge of GOQDs would inhibit the non-radiative decay, thereby affecting its optical properties [62].

### 3.5. Other Factors

In addition to the influence of temperature, pH, size, elements, and oxygen-containing functional groups, the PL of GO, GOQDs, and GQDs is also affected by some unconventional factors, such as metal ions, ion irradiation, and solvent induction. Wang et al. first studied the metal enhanced fluorescence (MEF) of GO sheets [63]. Compared with the glass substrate, the fluorescence intensity of GO on the silver substrate was enhanced by about 10 times. It is different from other fluorescent materials in that the direct contact between GO and metal exhibits a strong MEF, which indicates that GO sheets could potentially be used as fluorescent probes for three-dimensional optical imaging and sensing. In addition, they also studied the effects of other metal ions on the PL of GO. K^+^, Ca^2+^, and Mg^2+^ cause the increase of the fluorescence intensity of GO, while most transition metal ions except Zn^2+^ decrease the fluorescence intensity of GO. Wang et al. did not provide a clear mechanism on the reasons for the enhancement of PL. Nevertheless, they speculated that this may be due to the interaction between GO and metal ions with different electronic configurations and ionic radii, which led to different changes in the structure of GO.

Jayalakshmi et al. found that ion irradiation can also change its optical properties [64]. They obtained GO sheets by controlling the reduction method with 500 keV Ar^+^ ion irradiation, and measured their PL properties. In the process of ion irradiation, the content of sp^2^ hybrid carbon atoms in the sp^3^ matrix gradually increased with the Ar^+^ flux, and the PL and electrical properties of GO can be adjusted by changing the energy gap. The PL spectra of GO and rGO reduced by ion beam are shown in Figure 11a. Neogi et al. also studied the effect of ion irradiation on the PL of GO [65]. The PL emission of GO at different energies from visible to near-infrared regions showed that the existence of sp^2^-rich clusters with different sizes in the same sp^3^ substrate was a key factor to affecting PL.

Chen et al. studied the solvent-induced enhancement of optical properties of GO [66], and found that the PLQY of GO was 2.8% in ethanol, but 1.2% in water. Figure 11b displayed the steady-state fluorescence spectrum of GO in mixed solutions. The Raman spectrum signal of GO showed that the average size of polycyclic aromatic hydrocarbons of GO in ethanol was smaller than that of GO in water. The PLQY was higher as the solvent polarity was stronger. The interaction between chromophores and solvent molecules was stronger, and the microscopic scale movements of molecules promoted the non-radiative relaxation. These results showed that the interaction between GO and solvent molecules could reduce the PL intensity.

In addition, Jeon et al. found that the PL of GQDs could be regulated by the charge transfer effect of functional groups [67]. It was observed that the PL of GQDs shifted due to the charge transfer between functional groups and GQDs. Firstly, GQDs (GQDs-NHR) with amino functionalization, 1~3 layers in thickness, and less than 5 nm in diameter were successfully prepared. Figure 11c shows the PL images of GQDs (left) and GQDs-NHR (right) under the excitation of a 355 nm laser, which was compared with the unfunctionalized GQDs. Functionalized GQD exhibits a red shift (about 30 nm) due to the charge transfer between functional groups and GQD.

GO will appear to have serious aggregation and accumulation due to the size effect and surface effect, which will weaken the luminescence properties and reduce the PLQY. However, the fluorescence properties can be greatly improved after passivation treatment. Surface passivators can significantly improve the above phenomenon, and improve the luminescence properties of PLQY. Xiang et al. studied the effect of passivators with different molecular weights on the luminescence properties of GQDs [68]. The results showed that the surface of quantum dots passivated by large molecular weight polymers has long-chain functional groups, which makes the luminescence of quantum dots enhanced, and it can be seen from Figure 11d that with the increasing molecular weight of PEG, the PL intensity was significantly enhanced.

In addition, the external electric field can also lead to the change of PL properties of GO, as well as the fluorescence quenching of GO films. The effect of external electric field on PL and quenching will be discussed in Section 5.1.

## 4. Luminescence Characterization Methods

Spectral technology plays an irreplaceable role in studying element composition and energy level of matter. Various spectral technologies are often used to study the PL properties of GO. The most widely used luminescence characterization instruments include fluorescence microscope and fluorescence spectrometer. The fluorescence microscope is an instrument that uses a certain excitation light as the light source on the object, in order to emit fluorescence and observe its morphology and position under the microscope. With the development of characterization techniques, there are also high-resolution laser confocal fluorescence microscopes and full-spectrum laser confocal fluorescence microscopes. A fluorescence spectrometer is an instrument that can obtain much optical information such as the excitation spectrum, emission spectrum, quantum yield, and fluorescence intensity of matter.

Among the many spectral technologies, transient spectroscopy, which can study the exciton behavior of GO at ultra-fast time scales, is one of the most widely used methods for studying materials According to different measurement parameters, transient spectroscopy can be divided into transient absorption spectroscopy, ultra-fast infrared absorption spectroscopy, time-resolved Raman spectroscopy, and time-resolved fluorescence spectroscopy.

Time-resolved photoluminescence (TRPL) spectroscopy is the preferred tool to study the rapid electron inactivation process leading to photon emission, by detecting the dynamic process of the excited state radiation transition spectrum of matter over time under the irradiation of pulsed monochromatic light (Figure 12a). As early as 1969, Heim studied the optical properties of some materials through TRPL [69]. This technology has been often used to study the PL process and carrier lifetime of GO and other materials during the past decades [70]. As mentioned in Section 2.2, Chien et al. investigated the multi-exponential decay kinetics of GO in water from 1 ps to 2 ns through TRPL, and the specific process involves a pump light emitted by the laser that is reflected in the microscope system through a 2D color mirror. The fluorescence emitted by photogenerated carriers is collected through the objective and then passed through the microscope system. The reflected pump light is filtered through a 2D color mirror and focused on the spectrometer. The spectrometer has two outlets, which are connected to the spectrometer camera and the single-photon counter, respectively. The spectrometer camera is used to measure the steady-state fluorescence spectrum, while the single-photon counter is used to measure the time-resolved fluorescence spectrum [71]. Wang and co-workers found that that the fluorescence of GO (red band) is mainly due to the contribution of different sizes of sp^2^ regions via TRPL test [72].

The structure of ultrafast infrared absorption spectroscopy is similar to that of transient absorption spectroscopy, but their principles are slightly different (Figure 12b,c). In the transient absorption spectroscopy, the time-domain response of the probe’s optical signal is caused by excitons. The time-domain response of the probe’s optical signal of ultrafast infrared absorption spectroscopy corresponds to the concentration change of the electron–hole pair in the material. Wang carried out a detailed study on the migration/redistribution of different electron states in GO through ultrafast transient spectroscopy [72]. The results showed that a charge transfer state is formed between the sp^3^ hybrid carbon atoms and oxygen-containing functional groups in the sp^3^ region of GO. The energy and charge transfer process occurs in the time scale from sub-picoseconds to dozens of picoseconds, and this excitation energy transfer process does not occur in the sp^2^ region of GO, which indicates that the sp^2^ electronic state is very localized. Sun et al. analyzed the relation between the electronic structure and oxygen content quantitatively, by combining the results of transient absorption spectroscopy and XPS. In the transient spectrum, the insulator–semiconductor–semimetal transitions in GO and its reduced derivatives were directly observed, and it was found that these transitions initially occurred in a relatively narrow oxygen content range [73].

## 5. Fluorescence Quenching Mechanism and Regulation

Fluorescence quenching, which is a significant tool for the study of the assembly of macromolecules and interactions between substances, refers to the phenomenon whereby some substances reduce the fluorescence intensity of the original substances with fluorescence characteristics. GO and GOQDs can be quenched by other substances, and they can also strongly inhibit the emission of dye molecules. In some optical sensing applications, GO and rGO can be quenched by energy transfer mechanism and thus effectively used as the basis of biosensors, disease detection, and so on. In this section, the fluorescence quenching and regulation of GO will be discussed in detail.

### 5.1. PL Quenching of GO and GOQDs

The intrinsic quenching of GO and GOQDs under certain conditions has been reported in much of literature. Lee et al. developed a physical way to change optical and electronic properties of GO through an electric field to promote and control the application of optoelectronic devices, in order to reversibly fine-tune the photoelectric behavior of GO without changing its chemical structure [33]. In their work, field-related GO emission was studied in a film with a 1 mm thickness of GO/PVP (polyvinylpyrrolidone), and the fluorescence intensity decreased by 6% under an electric field of 1.6 V·μm^−1^ (Figure 13a). Interestingly, the fluorescence intensity gradually increased with the removal of the electric field, which means that it is a reversible process. When a higher electric field intensity was applied at the scale of a single GO sheet, it was found that most of the sheets had a fluorescence quenching phenomenon. Lee et al. explained it with a theoretical modeling of the exciton phenomenon in a GO sheet, and deemed that the separation of electrons, holes, and the electric field might reduce their recombination probability, thereby leading to GO quenching.

Chen et al. studied the PL quenching of GO with metal ions in an aqueous solution using the Stern–Volmer equation [74]. The results showed that the overall trend of the quenching efficiency is Fe^2+^ > Co^2+^ > Ni^2+^ > Cd^2+^ > Hg^2+^. The PL spectra of GO solution with different metal ions at a fixed concentration (10^−5^ M) are shown in Figure 13b. Furthermore, the steady-state and time-resolved PL spectra of the GO solution showed that the PL quenching is related to the non-radiative optical transition from the bridging state, which is caused by the hybridization of the sp^3^ orbit of GO and the 3d orbit of metal ions, proven by the density functional theory. Wongrerkdee et al. used the fluorescence quenching probe based on GQDs to detect metal compounds, and studied the effect of metal compounds on the fluorescence quenching efficiency of quantum dots [75]. The fluorescence quenching of quantum dots can be explained by non-radiative relaxation. The fluorescence quenching efficiency is proportional to the amount of copper acetate. Figure 13c,d showed the fluorescence spectra and Stern–Volmer diagrams of GQDs with different concentrations of copper acetate (0~2.5 mM). In order to explore the quenching mechanism, Wongrerkdee et al. calculated the electrochemical potential gap (ΔE). In the case of and without UV irradiation, the ΔE of GQDs mixed with copper acetate was 0.152 V and 0.104 V, respectively. The ΔE after UV irradiation is higher than that without UV irradiation, indicating that the electron transfer rate from GQDs (electron donor) to copper acetate (electron acceptor) is higher. These results can further explore the new fluorescent probes based on GQDs. In summary, this method can be more widely used in water resources and the detection of copper ions in water based on GQDs fluorescent quenching probes.

Volkova et al. studied the quenching of graphene suspension (GS) PL by saturated hydrocarbons [76]. For instance, the hydrocarbon C_8_H_18_ was attached to the edge of the composite graphene sheet through the presence of functional oxygen−containing groups. It is speculated that the quenching of PL in the formed structure is related to the Anderson electron density delocalization effect. The interaction between hydrocarbon molecules and the donor–acceptor pair of graphene defects can transform the radiation PL into non-radiation PL. Figure 13e showed PL spectra of GS excited at 270 nm.

Srivastava et al. studied the efficient fluorescence quenching of chemically exfoliated reduced GO [77]. They compared the fluorescence quenching between different graphite systems, such as rGO, GO, and graphite for the first time. Compared with graphite and GO sheets, rGO showed stronger quenching ability, because it had a larger surface area and more efficient π-π stacking. The spectral results (Figure 13f) showed that the quenching effect of chemically exfoliated rGO was nearly 16 times better than that of graphite and GO, by using rhodamine B dye as a quenching agent.

### 5.2. The Quenching of Other Substances by GO

It is worth noting that although GO itself is fluorescent, GO and the sp^2^ carbon network domain also allow the quenching of nearby fluorescent species such as dyes, conjugated polymers, and quantum dots [78]. Hamzah et al. revealed the quenching photoluminescence of silver nanoparticles (AgNPs) by GO sheets [79]. The excited electrons of Ag were attracted by the charged layer of GO, and this led to a lower radiative recombination of electrons and holes and eventually quenched fluorescence. Considering the efficient fluorescence quenching of GO on AgNPs, it is expected that the photoinduced electron transfer from AgNPs to GO will play an important role in quenching-related applications.

The quenching of dye molecules by GO has been studied extensively. Povedailo studied the fluorescence quenching of cationic dyes (acridine orange (AO), rhodamine B (Rd B), rhodamine 110 (Rd 110), and rhodamine 640 (Rd 640)) during the reaction with GO [80]. These cationic dyes can be used to track biological molecules in medical application. In addition, the PL of buffer solutions with different GO contents (pH = 6.0) showed that GO led to different degrees of PL quenching of these dyes according to the PL spectrogram. Zhao et al. ingeniously designed a novel nanostructure with a rigid adjustable silica interlayer to study the fluorescence quenching performance of GO in detail [81]. The distance between GO and fluorophore was controlled by adjustable silica interlayer. The results showed that even when the distance between the dye TAMRA and GO increases to more than 30 nm, the quenching efficiency of GO is still about 30% (Figure 14), indicating that GO has a long−distance quenching ability.

Unarunotai et al. studied that the fluorescence quenching efficiency of GO on dyes was affected by pH [82]. Different dyes were attributed to different quenching mechanisms (such as electrostatic interaction, π-π stacking, and hydrogen bonding). Affected by these mechanisms, the amount of sp^2^ clusters in GO sheets was further changed, thereby affecting the quenching efficiency. Lu et al. found that the PL of GOQDs was easily quenched by metal ions, and it was found that this quenching effect enhanced with the increase in pH values, which would introduce −OH to GOQDs, because −OH content increased under high pH conditions [83]. Therefore, the pH−dependent fluorescence quenching of GOQDs can be obtained, which can further guide the metal ion detection of GOQDs.

## 6. Applications of GO, GQDs, and GOQDs Luminescence

GO has been applied in many fields [84], such as fluorescence imaging [16], luminescence carriers [85], sensing matrices [20], biosensing [86,87], disease detection [88], drug carriers [89], ion detection [90], fluorescent probing [91], etc., due to its excellent PL properties. In this section, the PL of GO, GQDs, and GOQDs in electronic devices and biomedical fields will be emphatically introduced.

### 6.1. Electron Device

Although GO shows low electrical conductivity and structural stability, it has an energy band gap that can emit fluorescence in the visible–near-infrared region, which promotes a large number of photoelectric applications. Park et al. prepared GQDs by extracting microcrystals from an amorphic matrix of GO sheets, and this method has excellent characteristics such as large-scale production and long-term optical stability (Figure 15). Therefore, it has potential applications for PL fibers or films [92]. GOQDs with narrow size distribution were synthesized under different temperature conditions, but they have different PL properties, which also illustrates that there are other factors affecting the luminescence of quantum dots together with the effect of size.

Yin prepared green and red GQDs with optical properties; measured and analyzed their morphology, structure, and PL; and applied them to the preparation of white light-emitting diodes (WLEDs) [93]. In the driving current range of 20 mA to 350 mA, WLEDs showed excellent luminescent properties, indicating that WLEDs had great potential in the development of high-power devices. Shi et al. demonstrated the application of solution-processed GO as a hole injection layer in organic light-emitting diodes (OLEDs) [94]. By boosting the electron injection tactics and proper device packaging, brighter OLEDs with GO interlayer can be obtained. The research results represent that a cheap and flexible indium-free tin oxide electrode system can be developed by using GO materials in flexible OLEDs and other plastic electronic products.

### 6.2. Biomedicine

Graphene-based materials have been applied in the biomedical field for a long time and have achieved fruitful results. For instance, Potsi et al. developed a new strategy to obtain intrinsic PL graphene derivatives based on amine-functionalized fluorescent graphene [95]. The toxicity test using this strategy shows that hexamethylene diamine functionalized fluorographene (CHDMA) has excellent biocompatibility. Its clear green fluorescence can prove that these sheets are located in lysosomes of healthy cells (Figure 16a), which indicates that amine-functionalized graphene has real application potential in the fields of biological imaging, biosensing, and biomedicine. The application of GO in the fields of optical imaging and medical treatment has attracted great attention in recent years. In 2013, Li et al. reviewed the application of GO in this field in detail [96]. Therefore, only the biomedical application of GO after 2013 is introduced in this review.

In the field of fluorescence imaging, the PL characteristics of GO are applied to fluorescent probes for the purpose of fluorescence imaging. The following two basic criterions should be satisfied for fluorescent probes: first, they have to be easy to excite and have a high PLQY. To achieve biological imaging, PL characteristics are essential, and non-toxicity and certain targeting are also indispensable necessary conditions. The preparation method mainly determines whether it is non-toxic and green [16]. Sahoo et al. synthesized a green, low-cost, and environmentally friendly potassium-doped GO, and the potassium-doped content was as high as 6.81% [97]. Figure 16b displayed the confocal microscopic image of cells with potassium-doped GO as the fluorescent probe, which shows that the doped GO showed bright blue PL under the UV excitation of 365 nm, and its excellent optical properties help it to become an excellent biological imaging agent. When the non-neoplastic epithelial ovarian IOSE-364 cells were cultured with potassium-doped GO for 4 h, the water-soluble potassium-doped GO exhibits stable blue fluorescence properties.

Sastikumar et al. successfully synthesized non-toxic fluorescent NGOs by laser ablation of graphene solution and applied them to the field of biological imaging [98]. GO synthesized at varied concentrations (1, 5 and 10% *v*/*v*) had no direct cytotoxicity on smooth muscle cells at different moments. In addition, they did not affect the proliferation rate of smooth muscle cells. Figure 16c shows the confocal microscopy images of human vascular smooth muscle cells cultured for 6 h under the excitation of two different laser wavelengths (405 nm and 488 nm), and uniformly distributed bright blue fluorescence can be observed in the nucleus.

Carbon nanotubes (CNT) are derivatives of graphene materials, which have a high specific surface area, high mechanical strength, and high drug loading capacity. In recent years, CNT has been a popular material for drug delivery. However, CNT lacks biodegradability and has a certain toxicity, even though it has many excellent properties [99]. In contrast, GO does not only have the advantage of negligible toxicity, but also provides feedback on drug loading and release due to its optical properties [100,101].

Tang et al. have developed a GO-based DNA nanomaterial for cancer diagnosis and treatment in vitro and in vivo [102]. This material can be used for organ imaging in vivo, especially for liver tumors (Figure 17a). It has good biocompatibility, high detection specificity, and effective anti-tumor efficiency for target liver tumor cells. Liu et al. reported a new method to synthesize nanoscale water-soluble fluorinated graphene oxide (FGO) sheets, which has bright fluorescence (strong in acidic and alkaline conditions), high near-infrared absorption, and pH-responsive drug delivery capacity [103]. Through the interaction between FGO and drug fluorescence resonance energy transfer (FRET), it can be switched to luminescence for monitoring drug loading and release (Figure 17b).

## 7. Application of GO Fluorescence Quenching

### 7.1. Biomedicine

GO has been recognized as an effective fluorescent quenching material, which can reduce the brightness of the fluorophore [104]. This seemingly contradictory phenomenon is the performance of the atomic and electronic structure of GO. Because of the large conjugated structure of GO, it becomes an excellent electron acceptor in the process of energy transfer and then quenches the fluorescence [105]. The dual role of GO and GOQDs as fluorophores and quenchers makes them convert fluorescence signals dependent on the excitation wavelength and quench the fluorescence from external fluorophores, which shows great significance for the detection of various substances on a single biosensor [106,107].

Teniou et al. proposed a fluorescent aptamer sensor based on GO as a quenching agent and FRET principle for the rapid determination of dopamine (DA) [108]. This sensor showed excellent selectivity and sensitivity. In addition, the applicability of the sensor was confirmed by detecting DA in complex biological matrix, and the sensor had obvious accuracy.

Morales-Narváez et al. realized a real-time PL biosensor based on graphene oxide-coated microporous plates (GOMs) and PL biological probes (PLBs), which can be used as a rapid detection platform for pathogens [109]. Figure 18a,b show the preparation process diagram of GOMs and the principle of bacterial detection. GOMs deactivated the PL of PLBs without immune response through non-radiative energy transfer (A). PLBs that undergo immune responses (via antibody-bacterial membrane affinity) maintain their PL (B). PLBs that react with target bacteria in the liquid phase are not inactivated by GOMs for two reasons: (1) the distance between complexes (PLBs–bacteria) and GOMs, and (2) low affinity between the same complex and GO. The technology they proposed was high universality because GO could quench different fluorophores and detect other pathogens by simply changing the antibodies involved.

Zhao et al. developed a convenient, low-cost, and high-sensitivity fluorescent aptamer sensor based on GO (GO-apt) for leukemia detection [110]. GO and aptamer was used as fluorescence quenching agents and targeting agents, respectively. In the absence of leukemia cells (CCRF-CEM), GO can interact with carboxy fluorescein-labeled Sgc8 aptamer (FAM-apt) to quench almost all fluorescence, and the detection signal is closed. Nevertheless, when the target cells exist, the aptamer actively targets the cells and falls off from GO. Therefore, the concentration of target cells can be measured according to the change of fluorescence intensity. Figure 18c is the GO-apt fluorescent aptamer sensor diagram for CCRF-CEM detection.

### 7.2. Materials Imaging

As mentioned in Section 5.2, GO and quantum dots are strong quenchers of dye molecules. Huang et al. made full use of the reverse strategy of this feature to apply it to fluorescence quenching microscopes (FQM) [111]. Traditional microscopes, such as optical microscopes (OM), scanning electron microscopes (SEM), and atomic force microscopes (AFM), need a certain substrate or variety of strict conditions when imaging thin sheet materials. The FQM can quickly image nanosheets on any substrate with high imaging quality. In other words, FQM can rely on the quenching properties of GO to observe the imaging of GO itself [112]. Generally, GO is deposited on the imaging substrate in advance, then the fluorescent dye is placed on the substrate and excited at an appropriate wavelength. Many details of the GO sheet can be observed by FQM, such as wrinkles, curls, and overlaps. The mechanism of FQM is shown in Figure 19a,b. Furthermore, it can be extended to other 2D materials with excellent quenching properties (such as MoS_2_, MoSe_2_, WS_2_, etc. [113]) to observe their structural characteristics via the reverse strategy of GO. Figure 19c compares the imaging of MoS_2_ under OM, AFM, and FQM.

### 7.3. Anti-Counterfeiting

Aggregation-induced luminescence (AIE) is a phenomenon discovered and reported by Tang and co-workers in 2001, which refers to luminescence promotion by using molecular aggregation [114]. Fruitful work on AIE in the field of luminescence has been achieved during the past two decades. For example, the ingenious combination of molecules with AIE effect and GO with quenching effect has been widely used in the field of anti-counterfeiting. Chen et al. used tetraphenylethylene (TPE) with AIE effect and GO with quenching property to make TPE@GO composites (Figure 20a), and applied them to anti-counterfeiting security and other fields [115]. Tetrahydrofuran (THF) is a good solvent for dissolving TPE, while TPS shows clustering in water. At an appropriate proportion (*V*_THF_ = 60%), the THF/H_2_O mixture can promote the formation of TPE nanoparticles, resulting in fluorescence emission. In the mixed solvent, the TPE is first dissolved in THF, and then TPE molecules aggregate into nanoparticles in H_2_O with the volatilization of THF. In this process, the formation of particles increases the distance between GO and TPE nanoparticles beyond the quenching region of GO and emit fluorescence. In addition, they found that the fluorescence was quenched again after pure THF was sprayed again, due to the uniform dispersion of TPE nanoparticles dissolved in THF. Finally, they used GO and TPE dispersions as invisible ink to draw patterns and decrypted or encrypted information by spraying THF/H_2_O mixture or pure THF, respectively (Figure 20b–e).

## 8. Summary and Prospects

Through the research and development of graphene materials in the past decades, the latest achievements of GO in various fields have emerged in an endless stream. The disordered state of internal functional groups of GO makes it also known as a disordered material. It obtains tunable optical properties because of the internal disorder, which makes it applicable potentially in electronic devices, biomedicine, sensors, and other fields. This review mainly introduces the mechanism and control factors of PL and quenching of GO, the commonly used characterization methods of GO optical properties, as well as the applications of GO in many areas.

The optical properties of GO and corresponding mechanisms are also constantly clear through the effort made by many researchers. However, there are still many problems in the study of optical properties of GO, which still need further exploration.

(1) Although the current reduction methods (such as chemical reduction, thermal reduction, etc.) are relatively mature, there are also shortcomings, as it is hard to control the degree of reduction, complex reduction process, high requirements, and possibly irreversible changes in the original excellent properties of GO. Such shortcomings are bound to limit the application of GO. Therefore, it is still necessary to develop efficient, controllable, and friendly reduction methods for GO.

(2) The luminescence mechanism of GO and quantum dots still needs to be further clarified. Although several current PL mechanisms are discussed in this paper, there is no clear conclusion in the mechanism research section. For instance, the π-π* transition of the sp^2^ cluster is one of the highest acceptable mechanisms; although Chhowalla et al. deemed that the PL of GO is closely related to the size of sp^2^, this theory cannot be applied to GOQDs because PL quantum dots with size-dependent and size-independent can be synthesized.

(3) GO has been applied in many fields, but there are few fields where the optical properties of GO are applied, at present mainly concentrated in biomedical fields (such as biosensors, drug delivery, disease detection, etc.). Therefore, the applications of GO optical properties in others field should be explored in the future.

## Figures and Tables

**Figure 1 nanomaterials-12-02444-f001:**
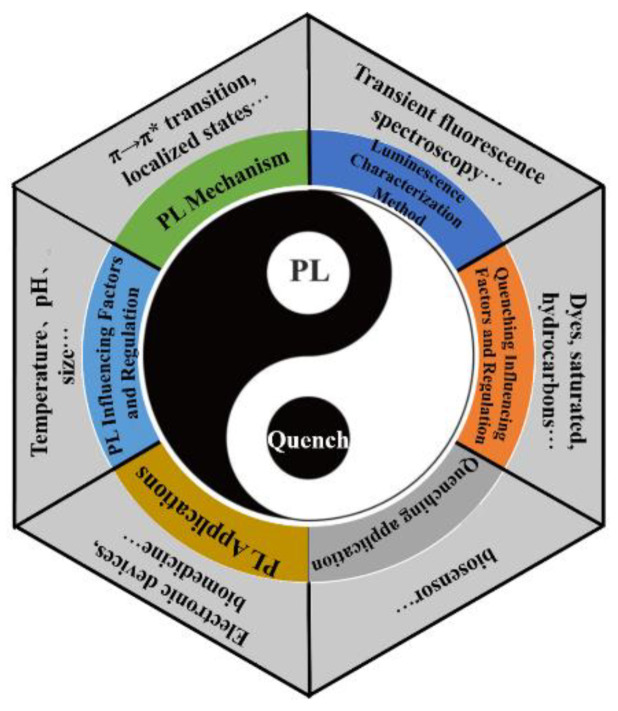
Research content of this review.

**Figure 2 nanomaterials-12-02444-f002:**
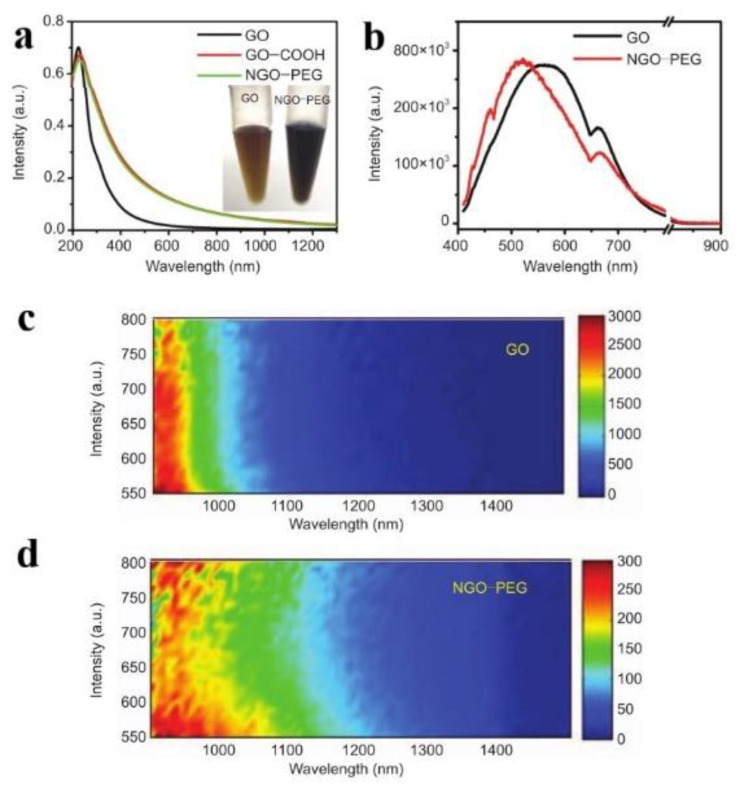
(**a**) Absorption spectra of GO, GO-COOH, and NGO-PEG solutions. (**b**) The fluorescence of GO and NGO-PEG in the visible region. (**c**,**d**) PL spectra of GO and NGO-PEG in the infrared region. Reprinted with permission from [17]. Copyright 2008, Springer Nature.

**Figure 3 nanomaterials-12-02444-f003:**
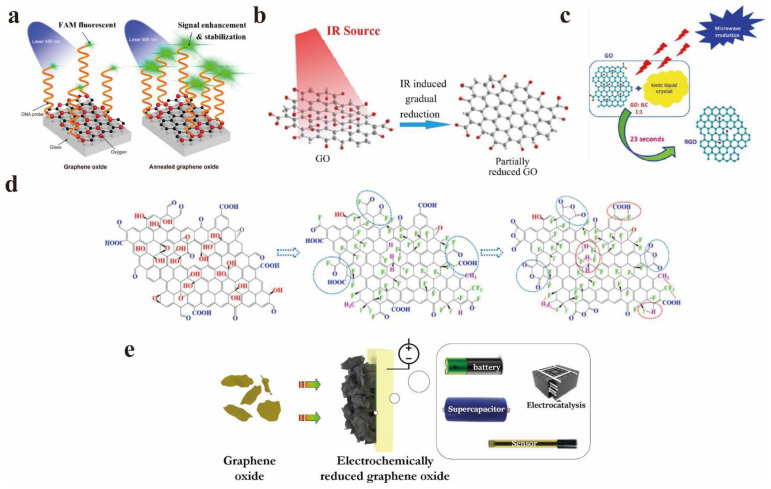
Summary of common reduction methods: (**a**) Annealing. Reprinted with permission from [20]. Copyright 2022, American Chemical Society. (**b**) Infrared radiation. Reprinted with permission from [24]. Copyright 2014, IOP Publishing Ltd. (**c**) Using the dielectric properties of ILC to promote rapid microwave heating. Reprinted with permission from [30]. Copyright 2022, Elsevier. (**d**) Fluorination method. Reprinted with permission from [23]. Copyright 2020, Elsevier. (**e**) Electrochemical reduction. Reprinted with permission from [31]. Copyright 2022, Elsevier.

**Figure 4 nanomaterials-12-02444-f004:**
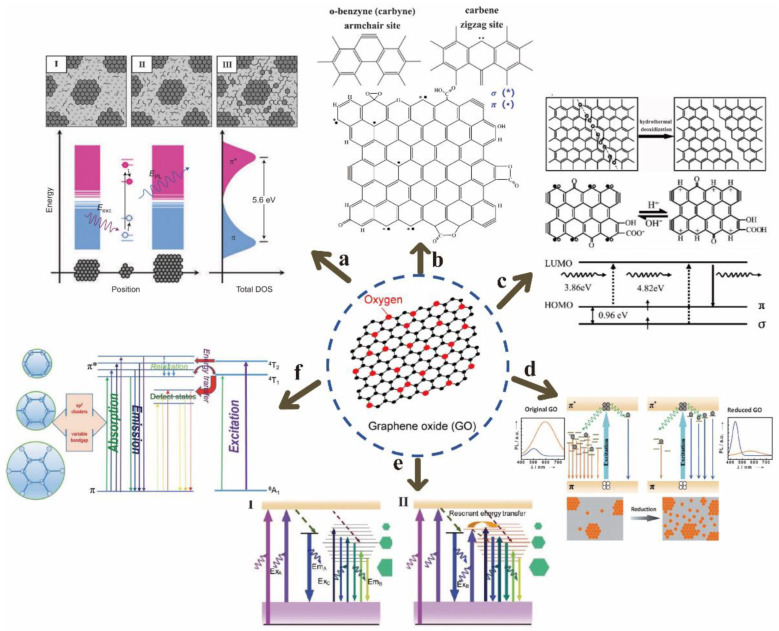
GO luminescence mechanism: (**a**) sp^2^ cluster π-π* transition. Reprinted with permission from [34]. Copyright 2010, John Wiley and Sons. (**b**) Free serrated (theoretical expectation). Reprinted with permission from [37]. Copyright 2005, John Wiley and Sons. (**c**) Free serrated (experimental confirmation). Reprinted with permission from [38]. Copyright 2010, American Chemical Society. (**d**) Disorder-induced localized state. Reprinted with permission from [39]. Copyright 2012, John Wiley and Sons. (**e**) Chemical reduction. Reprinted with permission from [40]. Copyright 2013, John Wiley and Sons. (**f**) Energy transfer process. Reprinted with permission from [41]. Copyright 2014, Royal Society of Chemistry.

**Figure 5 nanomaterials-12-02444-f005:**
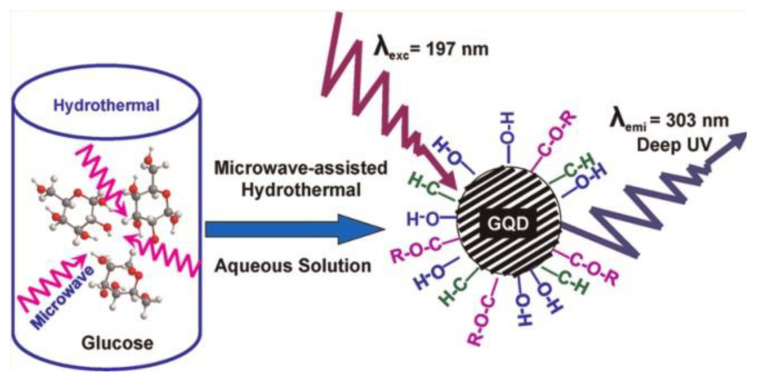
Self-passivation layer formed by surface states between π* band and π level. Reprinted with permission from [43]. Copyright 2012, American Chemical Society.

**Figure 6 nanomaterials-12-02444-f006:**
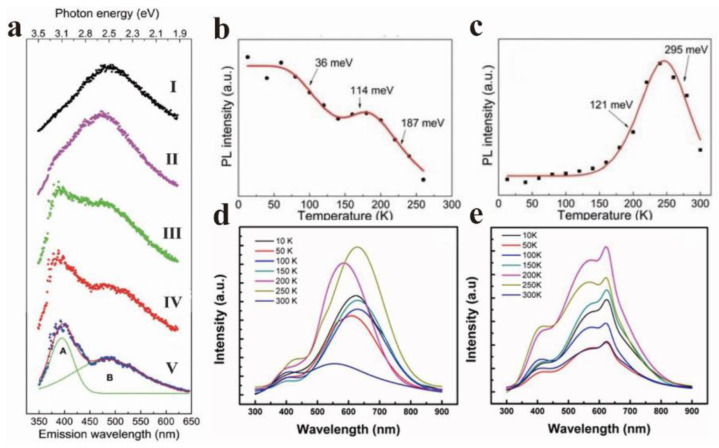
(**a**) PL spectra of GO deposited films at 13 K (I), 140 K (II), 180 K (III), 240 K (IV), and 300 K (V); (**b**) and (**c**) are the change of PL intensity of GO sheet in band A and band B with temperature. Reprinted with permission from [53]. Copyright 2014, Royal Society of Chemistry. (**d**) PL spectra of CRG and (**e**) TRG were obtained at temperatures from 10 K to 300 K. Reprinted with permission from [54]. Copyright 2011, AIP Publishing.

**Figure 7 nanomaterials-12-02444-f007:**
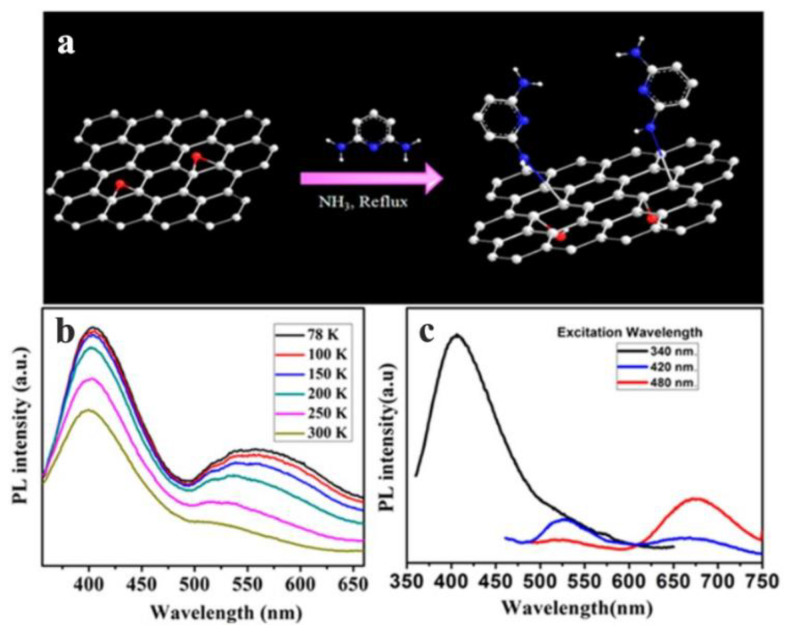
(**a**) Synthesis scheme of DAP-fGO. (**b**) Temperature-dependent and (**c**) excitation-dependent PL spectra of DAP-fGO films. Reprinted with permission from [55]. Copyright 2016, American Chemical Society.

**Figure 8 nanomaterials-12-02444-f008:**
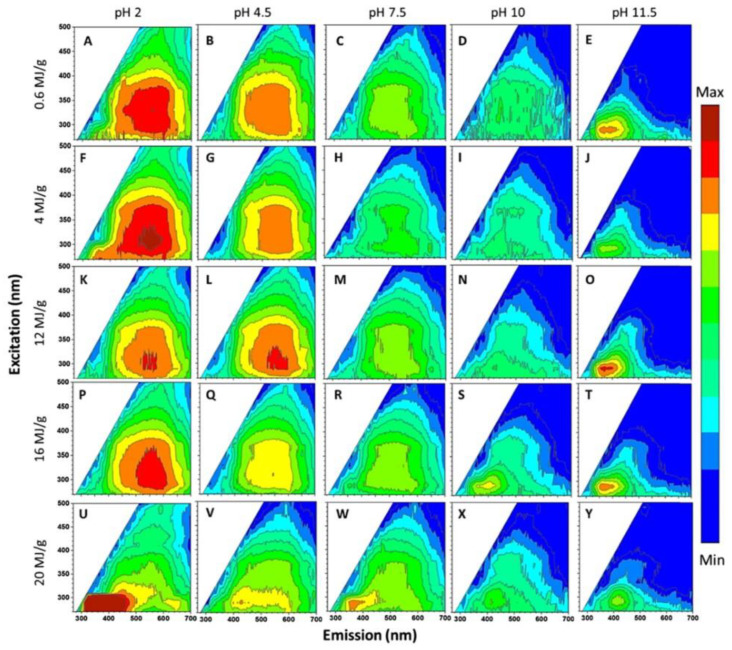
2D PL spectra of GO dispersions under different pH and ultrasonic conditions. Reprinted with permission from [57]. Copyright 2017, American Chemical Society.

**Figure 9 nanomaterials-12-02444-f009:**
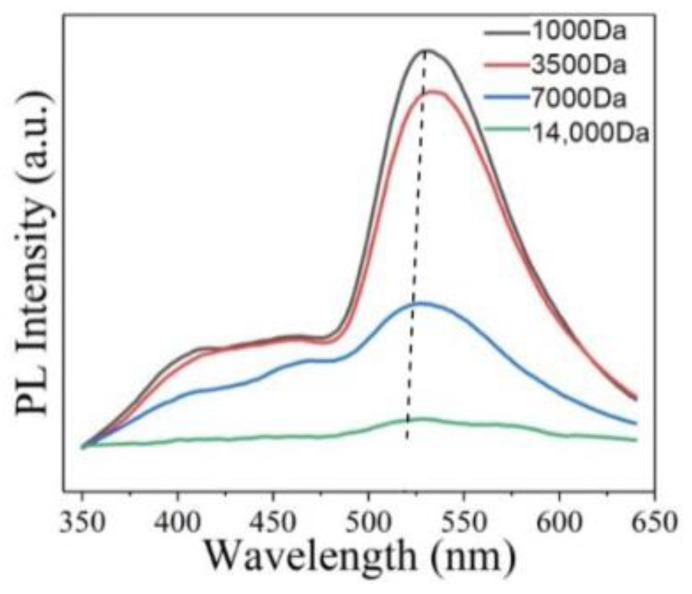
PL spectra of four separated quantum dots excited at 330 nm. Reprinted with permission from [50]. Copyright 2021, MDPI.

**Figure 10 nanomaterials-12-02444-f010:**
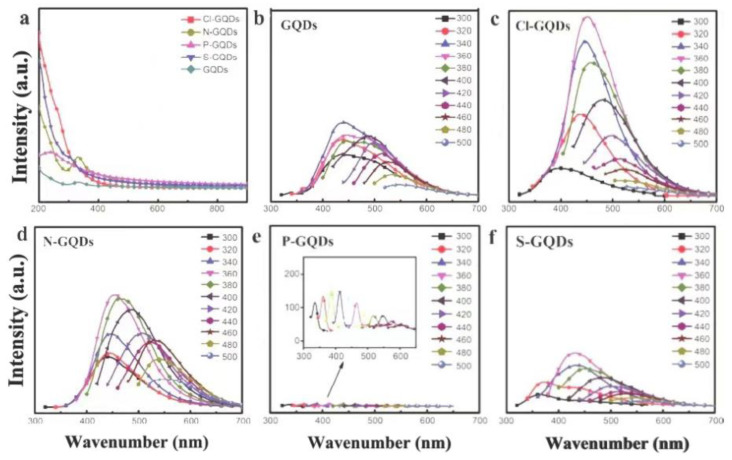
UV absorption spectra (**a**) and PL spectra (**b**–**f**) of GQDs and GQDs doped with different elements. Reprinted with permission from [51]. Copyright 2019, SCIENCE CHINA PRESS Co., Ltd.

**Figure 11 nanomaterials-12-02444-f011:**
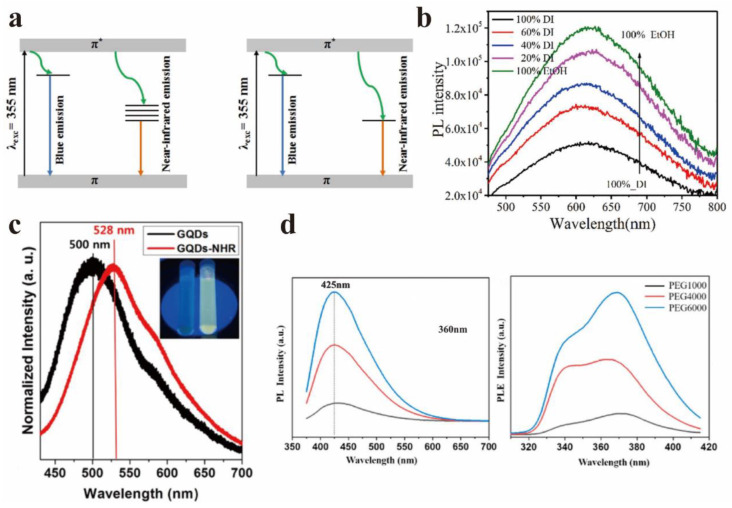
(**a**) Luminous band diagram of GO and ion beam reduction rGO. Reprinted with permission from [64]. Copyright 2018, IOP Publishing Ltd. (**b**) The steady-state fluorescence spectra of GO in mixed solution. Reprinted with permission from [66]. Copyright 2018, Elsevier. (**c**) GQDs (**left**) and GQDs-NHR (**right**) PL photographs excited by 355 nm laser. Reprinted with permission from [67]. Copyright 2013, American Chemical Society. (**d**) Effects of different molecular weight passivators on the luminescent properties of GQDs. Reprinted with permission from [68]. Copyright 2021, Elsevier.

**Figure 12 nanomaterials-12-02444-f012:**
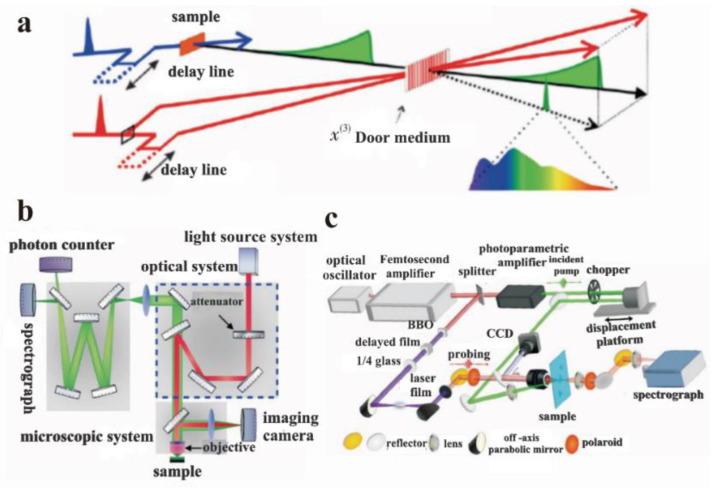
Structure diagram of partial spectral technique. (**a**) Transient fluorescence spectra. (**b**) Single-photon counting time-resolved fluorescence system. (**c**) Transmission transient absorption spectrum detection system. Adapted with permission from [71]. Copyright 2017, Electro-Optic Technology Application.

**Figure 13 nanomaterials-12-02444-f013:**
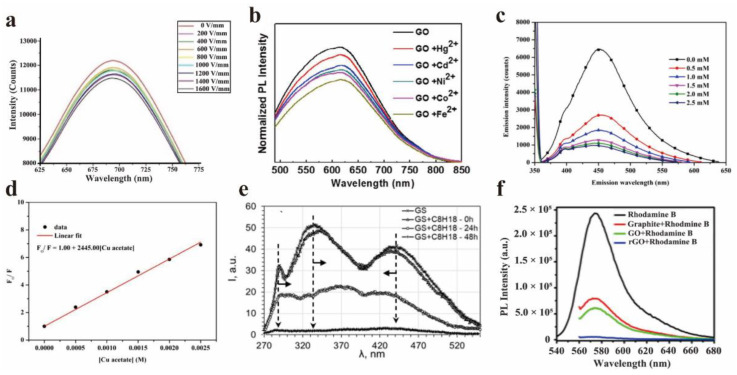
(**a**) PL spectra of GO in GO/PVP films under different electric field conditions. Reprinted with permission from [33]. Copyright 2020, IOP Publishing Ltd. (**b**) PL spectra of GO solution with different metal ions. Reprinted with permission from [74]. Copyright 2015, Elsevier. (**c**,**d**) The fluorescence spectrum and Stern–Volmer diagram of GQD with different copper acetate concentrations (0~2.5 mM). Reprinted with permission from [75]. Copyright 2021, Taylor & Francis, Web: www.tandfonline.com, accessed on 25 June 2022. (**e**) Effect of saturated hydrocarbon on photoluminescence spectra of GS (excited by 270 nm). Reprinted with permission from [76]. Copyright 2021, Elsevier. (**f**) Rhodamine B dye for graphene, GO, rGO fluorescence quenching PL. Reprinted with permission from [77]. Copyright 2009, American Vacuum Society.

**Figure 14 nanomaterials-12-02444-f014:**
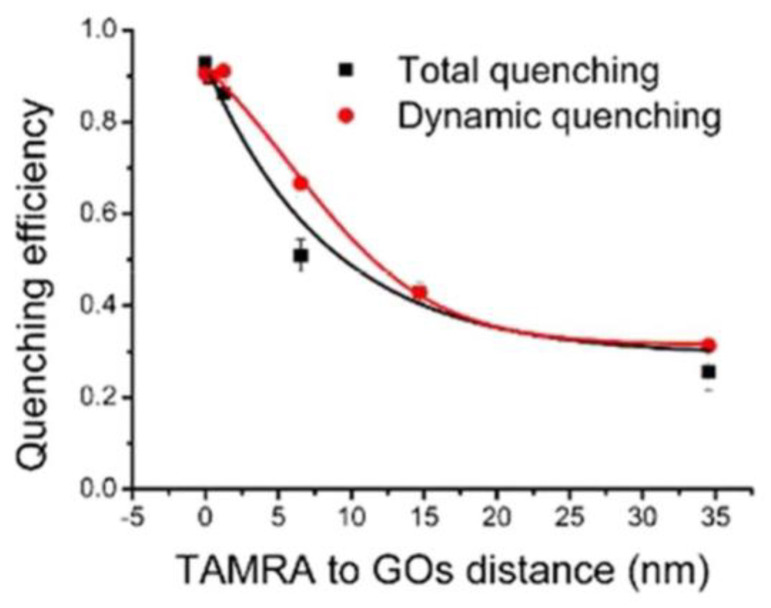
The effect of the distance between TAMRA and GO on the quenching efficiency of GO. Reprinted with permission from [81]. Copyright 2018, American Chemical Society.

**Figure 15 nanomaterials-12-02444-f015:**
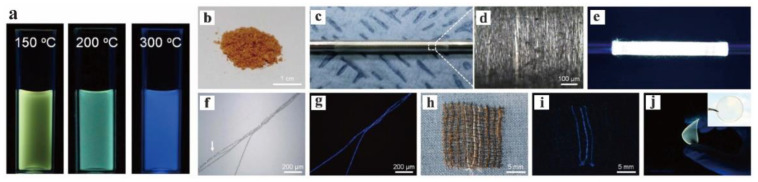
(**a**) Photographs of GQDs dispersions prepared at 150, 200, and 300 °C (excited at 365 nm). (**b**) Photos of freeze-dried GQD powder; the images of long GQD fibers were taken at (**c**) low magnification and (**d**) high magnification. (**e**) GQDs fibers emit strong yellow light excited at 365 nm. (**f**,**g**) Optical microscopic images of blue GQD/PAA composite fibers and PAA fibers under (**f**) bright-field conditions and (**g**) ultraviolet irradiation. (**h**) Photographs of three GQD yarns woven from commercial cotton fabrics under white and (**i**) ultraviolet light. (**j**) Photos of yellow GQD/PAA composite membrane excited at 365 nm and under white light (illustration). Reprinted with permission from [92]. Copyright 2015, Springer Nature.

**Figure 16 nanomaterials-12-02444-f016:**
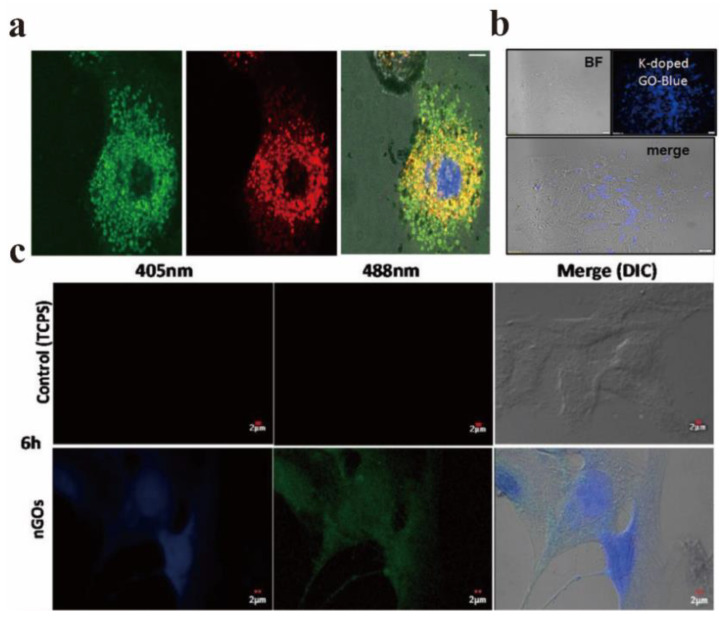
(**a**) Highly obvious co-localization of CHDMA (green pixels in the left image) and lysosome (red pixels in the middle image) in fibroblasts, showing the 3D reconstruction and merging of green and red channels in a plane layer of cells; the central blue circle is the position of the nucleus. Reprinted with permission from [95]. Copyright 2019, Elsevier. (**b**) Confocal microscopic images of cells with potassium-doped GO as fluorescence probe. Reprinted with permission from [97]. Copyright 2019, Elsevier. (**c**) Confocal microscopy images of human vascular smooth muscle cells cultured with 1% NGOs for 6 h. Reprinted with permission from [98]. Copyright 2020, Elsevier.

**Figure 17 nanomaterials-12-02444-f017:**
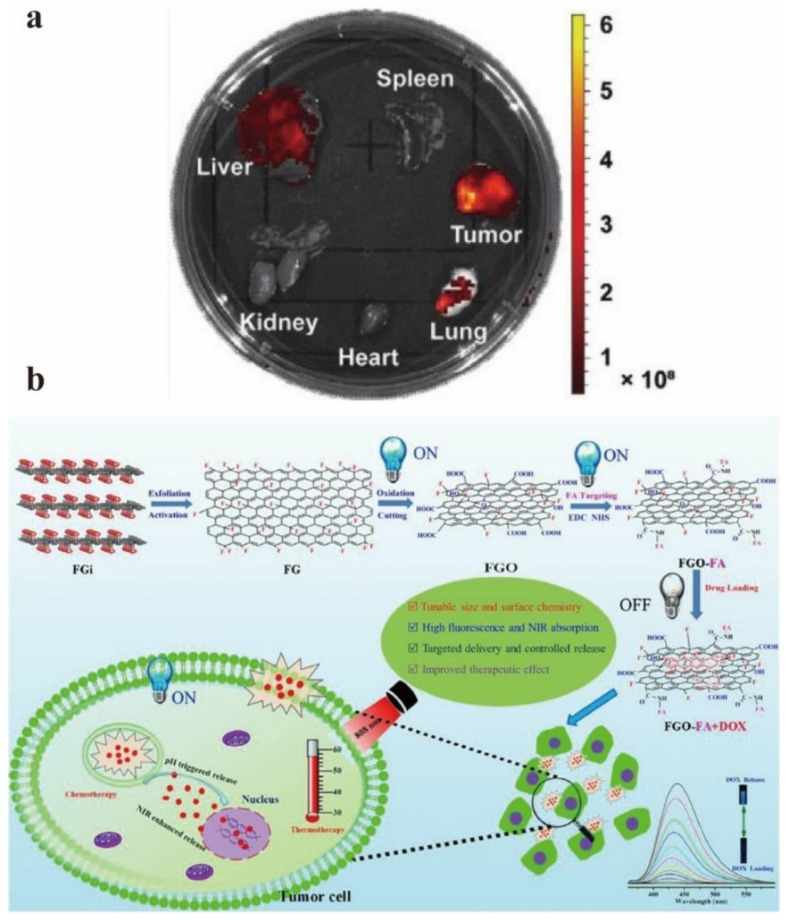
(**a**) Organ imaging 24 h after injection of GO–DNA nanomaterials. Reprinted with permission from [102]. Copyright 2021, John Wiley and Sons. (**b**) The schematic diagram of the targeted GO drug delivery system was prepared by controlling the structure and surface chemistry, convertible fluorescence and collaborative treatment. Reprinted with permission from [103]. Copyright 2018, Elsevier.

**Figure 18 nanomaterials-12-02444-f018:**
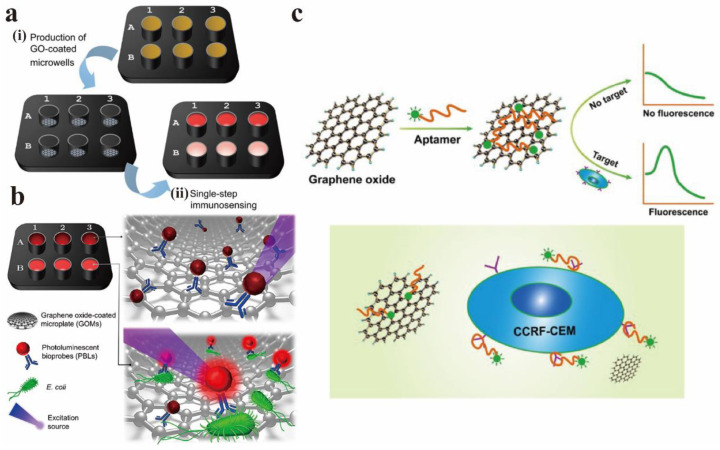
(**a**) GOMs preparation process diagram. (**b**) Principle of bacterial detection. Reprinted with permission from [109]. Copyright 2020, American Chemical Society. (**c**) The schematic diagram of fluorescent aptamer sensor for CCRF-CEM detection. Reprinted with permission from [110]. Copyright 2018, Springer Nature.

**Figure 19 nanomaterials-12-02444-f019:**
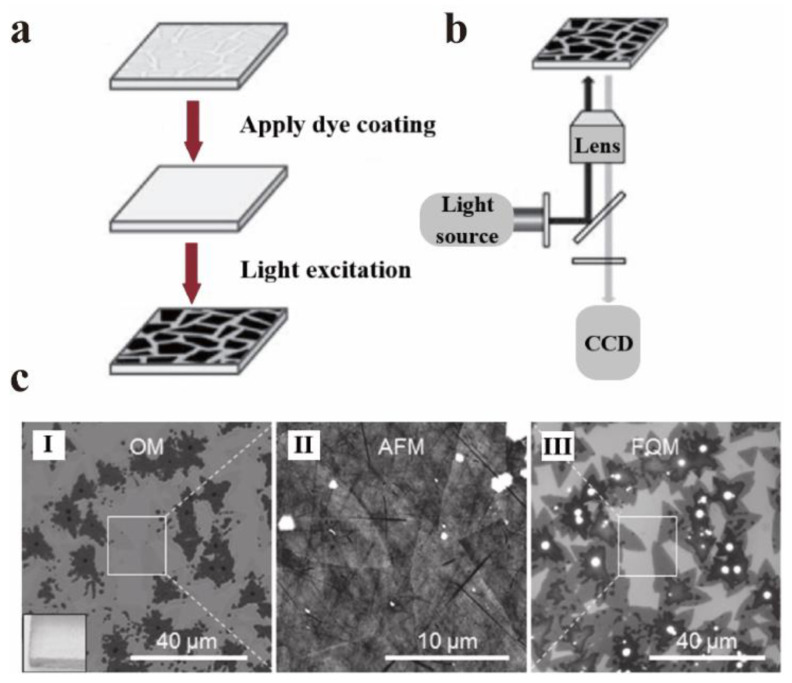
High throughput visualization of graphene substrate by FQM: (**a**) schematic diagram of fluorescent dye layer covering on GO substrate. (**b**) The schematic diagram of the substrate for the imaging of the lower sheet when excited. Reprinted with permission from [112]. Copyright 2014, Elsevier. (**c**) Comparison of images of OM (I), AFM (II), and FQM (III) using MoS_2_ as an example. Reprinted with permission from [113]. Copyright 2013, John Wiley and Sons.

**Figure 20 nanomaterials-12-02444-f020:**
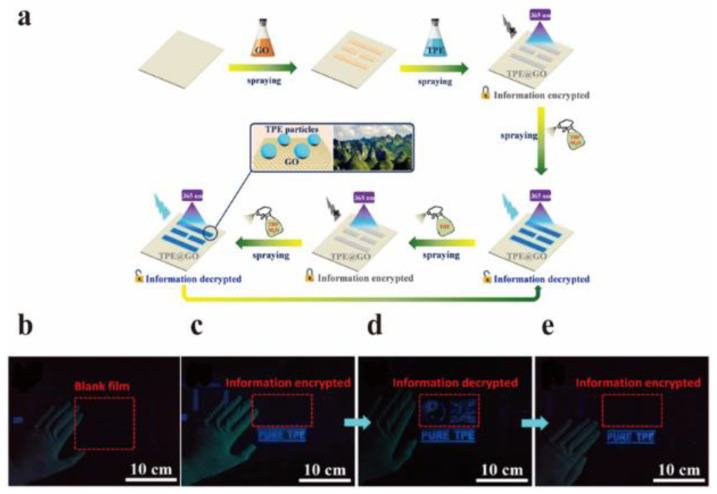
(**a**) The preparation TPE@GO invisible ink. (**b**–**e**) Under UV light: (**b**) blank group, (**c**) original spraying film, (**d**) THF/H_2_O mixture spraying film, and (**e**) optical images of THF/H_2_O mixture and pure THF spraying film in turn. Reprinted with permission from [115]. Copyright 2019, American Chemical Society.

## Data Availability

Not applicable.

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
