# Peer review of "Photoluminescence and Fluorescence Quenching of Graphene Oxide: A Review"

_nanomaterials, 2022, doi:10.3390/nano12142444_

Round 1
Reviewer 1 Report
The authors have done a good effort elaborating this review. The main point missing and that it is really important in this field is the definition of graphene oxide (GO) and reduced graphene oxide (rGO), i.e. the criteria for being GO and rGO according with the oxidation degree. They can check in papers working with GO and rGO in which authors provide the XPS elemental analysis O/C.
Author Response
Thank you very much for your valuable suggestions and affirmation of our work. We have made significant revisions and improvements to this paper based on your comments. In particular, the difference between GO and rGO you mentioned, as well as the relevant XPS data references, we have mentioned in the revised article and annotated with red pen (in Section 2.1 and Ref 18).

Reviewer 2 Report
Authors should include more references in many places, in Introduction and other sections,
Like "One of the major topics on GO is the photoluminescence (PL) and quenching of GO and its quantum dots (GOQDs) and graphene quantum dots (GQDs)."
"The PL is theoretically impossible to achieve for the graphene material due to the zero band gap" Need references".
Also, the author missed many important references that should be must include, like 1. DOI: 10.1002/adma.201606755;
2. DOI: 10.1039/c9bm01341e;
3. DOI: 10.1002/advs.201600217;
Author Response
Thank you very much for your valuable suggestions and affirmation of our work. According to your opinions, we have supplemented the relevant references in the corresponding part of the article (Ref. 10, 11, 12, 19). In addition, we seriously read the literature you recommended and agreed that this literature has a lot of enlightening significance for our current and future work. In addition, we also cited this article in the revision (Ref. 87, 91, 107). The above modification information is marked in red pen in the revised manuscript for you to consult again.
